

# A high Mn(II)-tolerance strain, *Bacillus thuringiensis* HM7, isolated from manganese ore and its biosorption characteristics

Huimin Huang[1], Yunlin Zhao[1], Zhenggang Xu[1], Yi Ding[1], Xiaomei Zhou[2] and Meng Dong[2]

[1] Hunan Research Center of Engineering Technology for Utilization of Environmental and Resources Plant, Central South University of Forestry and Technology, Changsha, Hunan, China
[2] School of Material and Chemical Engineering, Hunan City University, Yiyang, Hunan, China

## ABSTRACT

Microorganisms play a significant part in detoxifying and immobilizing excessive metals. The present research isolated a strain (HM7) with high Mn(II) tolerance from Mn(II)-contaminated soil samples. The 16S rDNA sequence analysis showed that HM7 had a 99% similarity to *Bacillus thuringiensis*, which can survive under a high concentration 4,000 mg/L of Mn(II), and the highest removal rate was up to 95.04% at the concentration of 400 mg/L. The highest Mn(II) removal rate was detected at the contact time 72 h, temperature 30 °C, and pH 5.0, while the differences in strain growth and Mn(II) removal rate among different inoculation doses were insignificant. Scanning electron microscopy indicated *B. thuringiensis* HM7 cells appeared irregular and cracked under Mn(II) stress. Fourier transform infrared exhibited that functional groups like carboxyl, hydroxyl, amino, sulfhydryl groups, and amide bands might take part in the complexation of Mn(II). In addition, HM7 suggested the ability of indoleacetic acid production, siderophore production, and P' solubilization potential. Therefore, HM7 might have a potential to promote metal absorption by changing the form of heavy metals, and the experiments supported the application of *B. thuringiensis* HM7 as a biological adsorbent in Mn(II) contaminated environment remediation.

## INTRODUCTION

In recent years, with the rapid development of industrialization, heavy metal pollution has become a great concern worldwide (*Li et al., 2014*). In particular, waste tailings contain high levels of toxic heavy metals, causing severe heavy metal pollution in the soil nearby, and harming human health through food chain enrichment (*Klimek, 2012*; *Zaidi et al., 2006*). Thus, the research on the restoration of heavy metal pollution in soil and water is imminent.

Soil is an important part of the earth's ecological environment, and the accumulation of heavy metals in polluted soil is not conducive to human life and health (*Huang et al., 2019*). Heavy metals in polluted soils are characterized by poor mobility, strong concealment,

Corresponding author
Zhenggang Xu, rssq198677@163.com

long retention period, and difficulty in degradation. They not only affect soil structure and nutrients, but also affect the surrounding environment and human health through infiltration, rainwater leaching and food chain (*Chen & Zhu, 1999*; *Kamaludeen et al., 2003*). Manganese (Mn) is an essential trace element in organisms, but excess Mn(II) causes a great damage to the metabolic activity of organisms. Moreover, the toxicity of Mn(II) has been confirmed that it may be the second limiting factor of acidic soil (*Mou et al., 2011*). In Mn ore, the concentration of Mn(II) is quite high, and as metals are difficult to be absorbed and degraded, they may enter the food chain in the environment, resulting in toxic, carcinogenic and mutagenic effects of organisms. Therefore, knowing how to repair the Mn-contaminated pollution has become an important issue to be solved in the current exhibition.

In order to solve metal pollution problems, many conventional methods of removal and detoxification of Mn(II) from contaminated environment have been reported such as ion exchange, chemical precipitation, electro dialysis, reverse osmosis, filtration, solvent extraction, and chemical oxidation–reduction (*Jarpa et al., 2016*). However, most of the traditional methods are inefficient, expensive, unsuitable and producing secondary pollutants such as toxic by-products (*Kang et al., 2015*; *Wang et al., 2017*). Therefore, it is imperative to find an effective and eco-friendly method with the detoxification and removal of Mn(II) for the safety of environment. As a safe, green, economical and simple treatment method, bioremediation has been widely used in contaminated soil management. It mainly includes three aspects of plant, animal and microbial restoration. Among them, microbial remediation mainly refers to the microbes in the environment to reduce the toxicity of heavy metals in the soil through the action of redox, precipitation and absorption of heavy metals (*Chanmugathas & Bollag, 1986*). Researches have shown that large amounts of microorganisms (fungi, bacteria and yeasts) play a significant role in bioremediation, they can change the form of the presence and reduce the toxicity of heavy metals by biosorption, oxidation–reduction reactions, and dissolving (*Glick, 2010*; *Nele et al. (2009)*; *Rajkumar et al., 2010*). *Hullo et al. (2001)* demonstrated *Bacillus subtilis* spore-forming colonies were darker at higher Mn(II) concentrations, which had spores that expressed Mn(II) oxidize. This was mainly because the CotA protein on *Bacillus subtilis* was involved in the metabolism in protection against hydrogen peroxide. *Lysinibacillus sp.* grown at the Mn(II) concentrations of 1 mM, the total Mn(II) removal rate reached 94.67% and 55.94% of Mn (II) was oxidized, this was because that most Mn(II) oxides in nature were formed through the life activities of microorganisms, microorganisms catalyzed the conversion of Mn(II) into oxides and improved the oxidation rate of Mn(II) (*Tang et al., 2016*). However, there are many factors influence the composition of microbial community. Researchers have detected that characteristics of environmental (e.g., metal concentration, temperature, pH, etc.) affected the cell biological activity and community (*Tuffin et al., 2006*; *Wang et al., 2011*).

Furthermore, microbes can not only directly repair the environment and remove heavy metals, but also can be combined with phytoremediation, mainly in the following aspects: (i) Microorganisms, such as mycorrhiza and endophytes, form a union with plant roots to enhance plant survival and growth rate by enhancing plant resistance( (*Moreira et al., 2019*);
(*Orlowska et al., 2011*). (ii) Microorganisms can optimize plant rhizosphere environment by converting heavy metal forms and improve plant survival conditions (*Kamaludeen et al., 2003*; *Rajkumar et al., 2010*). (iii) In synergy and symbiosis of plants and microorganisms, microorganisms metabolize and secrete plant growth-promoting substances, then increase plant root absorption, and the plant's ability to transport heavy metals (*Aboushanab et al., 2008*; *Compant, Clément & Sessitsch, 2010*). *Wani, Khan & Zaidi, (2007)* demonstrated that the *Bacillus sp*. had the promotion of plant growth, the adsorption of Zn and Pb, the production of siderophores and solubilization of insoluble phosphorus (P). Many studies have shown that microorganisms can not only adsorb heavy metals, but also promote plant transport of heavy metals and plant-growth (Table 1).

There is a valuable microbe bank in ore area with tremendous microbe resources. In the previous study, we isolated two high-efficiency Mn(II)-resistant bacteria from Xiangtan Mn(II) ore and analysed the tolerance characteristics of the strain. The present research was aimed to isolate Mn(II)-tolerant bacteria with the ability to promote plant growth from Mn(II)-polluted water. To gain more characteristic information on this Mn(II)-tolerant strain, relevant detects including 16S rDNA sequence analysis, physiological and biochemical test were performed to figure out the properties of the strain. Besides, the Mn(II) adsorption capacity of the tolerant strains and optimal biosorption conditions were detected through different parameters experiments. Scanning electron microscopy (SEM) analysis was carried out to reveal the cell surface changes in cell–surface after biosorption. Fourier transform infrared (FTIR) techniques were utilized to detect the functional groups in the strain surface. Finally, the property of plant growth promotion and the antibiotic resistance of the strain were tested. The main objective of the study was to provide an effective and environmentally friendly biosorbents which contribute to the bioremediation of heavy metal contaminated soils and water.

## MATERIAL AND METHODS

### Samples and main materials

Soil samples (the content of Mn(II) was about 15225.17 mg/kg, pH was about 4.5–4.7) were collected from Xiangtan Mn ore, Hunan, China (112°45′E∼122°55′E, 27°53′N∼28°03′N), which has serious heavy metal pollution (*Ouyang et al., 2016*). Samples were taken from 0–10 cm soil from Mn(II) ore wasteland, 100 mesh sieved, mixed evenly and placed in a sterile centrifugal tube, and stored at −80 °C in the laboratory.

Basic beef-Protein medium: Peptone 1%, beef extract 0.3%, and NaCl 0.5%, pH 5.6–6.0 (plus agar 1.5% for solid medium); Selection medium: $MnSO_4 \cdot H_2O$ was added to the basal medium and adjusted to the desired concentration. Required pH was adjusted by 1 mol/L NaOH and 1 mol/L HCl.

### Selection of Mn(II)-tolerant bacterial strains

Ten gram of soil sample was dissolved in 90 ml of distilled water and coated on 500 mg/L Mn(II) solid selection medium (basic beef-Protein medium), at 30 °C for 3 days cultured in a biochemical incubator (SPX-250B, China). In the aseptic console (SW-CJ-2F, China), the selected colonies were subsequent culturing/ sub-transferring incubated with increasing

Huang et al. (2020), *PeerJ*, DOI 10.7717/peerj.8589

**Table 1** Recent examples of phytoremediation promotion by microbes.

| Microorganisms | Plants | Mechanisms of microorganisms | Effect of microorganism on plant under the heavy metal pollution | References |
|---|---|---|---|---|
| *Bacillus licheniformis* | *Vigna radiata* | The bacteria produced indole acetic acid and had the ability of phosphate solubilisation. | The bacterium offered potential of plant growth promotion and the ability to Cu(II) (34.5%) and Zn(II) (54.4%) removal. | *Biswas et al. (2018)* |
| *Bacillus safensis* FO-036b, *Pseudomonas fluorescens* p.f.169 | *Helianthus annuus L.* | The bacterium produced the intensification of the rhizosphere process and changed EC, pH and dissolved organic carbon in the rhizosphere. | The treatments with inoculation reduced the exchangeable Pb (from 10.34% to 25.92%) and Zn (from 13.68% to 30.82%) in the rhizosphere. | *Mousavi et al. (2017)* |
| *Bacillus flexus* ASO-6 | *Oryza sativa* , | The strain showed high resistant to As (32 and 280 mM) and exhibiting elevated rates of As(III) oxidation. The ability of producing IAA, siderophore and to solubilize phosphate. | The bacterial inoculation increased grain yield, straw yield, and root biomass in the rice plant and As accumulation in straw and grain decreased significantly. | *Das et al. (2016)* |
| *Bacillus megaterium* 1Y31 | *Pennisetum* | The strain could produce IAA and effect the leaf differentially expressed proteins (e.g., photosynthesis, energy generation, metabolisms, and response to stimulus). | The strain increased the total Mn uptake (ranging from 23% to 112%) and dry weights (ranging from 28% to 94%) of pennisetum with treated compared to the control. | *Zhang et al. (2015)* |
| *Bacillus thuringiensis* KQBT-3 | *Tricholoma lobayensis* | The inoculating of KQBT-3 further induced oxidative response and alleviated lipid peroxidation in the plant due to Pb accumulation. | KQBT-3 could decrease malondialdehyde in the plant, biomass and accumulation of Pb increased 47.3% and 33.2%, respectively. | *Li et al. (2015)* |
| *Trichoderma* sp. PDR1-7 | *Pinus sylvestris* | PDR1-7 participated in the antioxidative defence process in the plants to facilitate nutrient uptake and by reducing heavy metal stress. | PDR1-7 promote plant growth and improve the absorption of As, Cd, Cu, Ni, Pb, and Zn. | *Babu, Shea & Oh (2014)* |
| *Pseudomonas* sp. Lk9 | *Solanum nigrum L.* | Lk9 could improve P and soil Fe mineral nutrition supplies and secrete organic acids to enhance soil heavy metal availability. | After inoculation, *S. nigrum* increased shoot dry biomass by 14% and the total of Cu by 16.0%, Zn by 16.4% and Cd by 46.6% accumulated in aerial parts. | *Chen et al. (2014)* |
| AMF | *Dendrocalamus strictus* | The strain could promote nutrient absorption in the plant (e.g., P, K, Ca, Mg, ect.). | The content of Fe and Al were reduced in plants. | *Babu and Reddy (2011)* |
| *Bacillus* sp. | *Brassica campestris L.* | The strain had high resistant to Cr, Zn, and Pb, and secreted organic acid. | The strain promote plant growth. | *Wani, Khan & Zaidi (2007)* |

concentrations of Mn(II) (500, 1,000, 1,500, 2,000, 3,000, and 4,000 mg/L) at 30 °C for 3 days in solid basic beef-Protein medium. Finally, the high Mn(II)-tolerant strain was picked up and repeatedly streaked on solid basal medium to obtain pure culture bacteria, stored at −4 °C in the Beef-Protein slant medium.

## Identification of the Mn(II)-tolerant strain
### Bacterial characterisation

By screening, a Mn(II)-tolerant strain was isolated and characterised morphologically, and biochemically. Gram staining, catalase test, hydrogen sulphide test, methyl red test, Voges Proskauer test, oxidase test, glucose test, and gelatine liquefaction test were detected by the standard methods in Bergey's Manual of Determinative Bacteriology (*Holt, 1994*).

### 16S rDNA-based identification

To determine the classification of the strain, 16S rDNA analysis was performed on HM7. The strain was subjected to PCR amplification using bacterial universal primers 1492R (5′TAC GGY TAC CTT GTT ACG ACT T 3′) and 27F (5′AGA GTT TGA TCM TGG CTC AG 3′). Purification and sequencing (Illumina Hiseq 2500, California, USA) were carried out by Shanghai Majorbio Technology Company, China (http://www.majorbio.com). The obtained nucleotide sequence data was deposited in the NCBI GeneBank sequence database, the online program BLAST was used to figure out the relevant sequences with known taxonomic information in the Genebank (http://www.ncbi.nlm.nih.gov/BLAST) to identify the strain. The phylogenetic tree was built by software Mega 7.0 (*Kumar, Stecher & Tamura, 2016*) which based on the neighbour-joining (*Saitou & Nei, 1987*).

### Biosorption tests

Experimental cultured conditions as follows, the concentration of Mn(II) was 1,000 mg/L, the cultured time was 72 h, the temperature was 30 °C, the inoculation biomass was one mL (with an approximate optical density of 1.0 at 600 nm, approximately $1.0 \times 10^{13}$ cells/L), the pH was 5.6–6.0. All tests were done in 250 ml conical Erlenmeyer flasks with 100 ml selection medium on a rotary shaker at 120 r/min (ZHWY-211C, China).

In order to detect the effect of different factors on the adsorption of $Mn^{2+}$ by the strain, one of the corresponding parameters were changed according to the above conditions. (i) Effect of the different initial Mn(II) concentrations: the cultured solutions were prepared containing $Mn^{2+}$ at concentrations of 0, 200, 400, 600, 800, 1,000, 2,000, 3,000, 4,000, 5,000, 7,500, and 10,000 mg/L with $MnSO_4 \cdot H_2O$. (ii) Effect of the cultured time: the contact time was investigated of 0, 12, 24, 36, 48, 60, 72, 84, 96, 108, 120, and 144 h. (iii) Effect of the temperature: the biosorption tests were studied at 5, 10, 15, 20, 25, 30, 35, and 40 °C. (iv) Effect of the inoculation biomass: the cultured solutions inoculated 0.1, 0.5, 0.5, 1.0, 2.0, and 5.0 mL of biomass separately. (v) Effect of the pH: the pH of the culture solution was adjusted to 3.0, 4.0, 5.0, 6.0, 7.0, 8.0, and 9.0 by using 1 mol/L HCl and 1 mol/L NaOH.

For the purpose of detecting the strain growth and Mn(II) removal rate in the tests, the samples were carried out at a specific time. The growth of HM7 was determined by ultraviolet spectrophotometer (UV-2450, China) at 600 nm ($OD_{600}$). The specific method

was preheating the ultraviolet spectrophotometer for 20 min before the experiment, and adjusting the wavelength to 600 nm. After the sample reaction was completed, 3-5mL of the bacteria suspension were taken into the quartz cuvette and placed it in the UV spectrophotometer to read the $OD_{600}$ value of the strain immediately. Than the bacteria suspension was centrifuged at 4,000 rpm for 15 min (TD5A, China), and the residual Mn(II) concentration of the supernatant was analyzed by flame atomic absorption spectrophotometer (AA7000, Japan). The pH was tested by Rex laboratory pH meter (PHS-3C, China). All experimental tests were repeated three times, data processing and variance analysis were determined by SPSS 20.0 (*Li & Chen, 2010*).

$$Q = \frac{(C_0 - C_e)}{C_0} \times 100\%$$

where Q is removal rate of Mn(II), $C_0$ is the initial concentration of Mn(II) (mg/L), and $C_e$ is the residual concentration of Mn(II) (mg/L).

### Scanning electron microscopy (SEM) analysis

SEM analysis (JSM-6380LV, Japan) was used for the purpose of observing the changes in surface morphology of cells before and after adsorption of $Mn^{2+}$ (under the conditions of initial Mn(II) concentration 1,000 mg/L, temperature 30 °C and pH 5.6–5.8 for 72 h). Mn(II)-loaded and Mn(II)-free strain samples were examined after cell fixation with glutaraldehyde, dehydrated with ethanol and freeze-drying with a vacuum.

### Fourier transform infrared (FTIR) analysis

In order to figure out the main chemical functional groups of $Mn^{2+}$ adsorbed by the cells, the strains before and after the adsorption of $Mn^{2+}$ were analyzed by FTIR (NICOLET 5700, USA). After culturing the strain under the conditions of initial Mn(II) concentration 1,000 mg/L, temperature 30 °C and pH 5.6–5.8 for 72 h in a liquid medium, the cells were collected by centrifugation and washed three times with deionized water, after vacuum freeze-drying, a small number of lyophilized cells were mixed with KBr, pressed at 10 $t/cm^2$ for 1 min, and determined by FTIR spectrometer (Wave number range of 400 $cm^{-1}$–4,000 $cm^{-1}$).

## Property of plant growth promoting and antibiotic resistance of the strain

To figure out the promoting potential of the strain, property of plant growth promoting and antibiotic resistance were checked. The test of Indole acetic acid (IAA) of the strain was detected by the method from *Sheng et al. (2008a)*. Mineral phosphate solubilization activity of the strain was determined in the Pikovskayas medium (*Biswas et al., 2018*). The siderophores production of the strain was detected by the CAS plate method (*Schwyn & Neilands, 1987*). The antibiotic resistance of the strain was tested in the Muller-Hinton agar medium by disk diffusion method (*Bharagava & Mishra, 2017*).

**Table 2  Morphological and biochemical characteristics of *B. thuringiensis* HM7.**

| Tests employed | Characteristics observed |
| --- | --- |
| **Morphology** | |
| Gram reaction | + |
| Shape | Short rod |
| Pigments | − |
| **Biochemical reactions** | |
| Catalase | + |
| Hydrogen sulfide | − |
| Methyl red | + |
| Voges Proskauer | + |
| Oxidase | + |
| Glucose | + |
| Gelatin | + |

**Notes.**

"+"and "−" indicate positive and negative reactions, respectively.

# RESULT AND ANALYSIS

## Selection of the optimal Mn(II)-resistant strain

In this study, a highly effective Mn(II)-tolerant strain HM7 was isolated from the soil of Mn ore by gradient acclimation culture. HM7 could tolerate 4,000 mg/L of Mn(II) concentration in screening tests and the removal rate of at low concentration (400 mg/L) reached up to 95%. The strain was used for further identification and analysis of Mn(II) removal properties.

## Characterization and molecular identification of HM7

According to morphological, biochemical and physiological tests, HM7 was identified as a rod-shaped bacterium. Hydrogen sulphide test was negative, while Gram-staining, catalase, methyl red, Voges Proskauer, oxidase, glucose, and gelatine tests were positive (Table 2).

HM7 was submitted to 16S rDNA gene sequence analysis for further confirm. The sequence of 16S rDNA of HM7 was subjected to GenBank (accession number MG787231). Searched for the similar strain with known DNA sequences on the National Center for Biotechnology Information (NCBI) by using the BLAST program (https://blast.ncbi.nlm.nih.gov/Blast.cgi). The results indicated that HM7 had a close genetic relatedness of *Bacillus* and 99% homology to *Bacillus thuringiensis* (*B. thuringiensis*) (ACNF01000156). The phylogenetic tree (neighbour-joining) was constructed by MEGA7 (*Kumar, Stecher & Tamura, 2016*) (Fig. 1). The higher identical value confirmed HM7 to be *B. thuringiensis* and was named *B. thuringiensis* HM7 in the study.

## Characteristics of biosorption under different culture conditions
### Effect of initial Mn(II) concentration on biosorption

The effect of initial Mn(II) concentration on biosorption by *B. thuringiensis* HM7 was carried out over a range of 0–10,000 mg/L. The results showed that the growth of HM7 within a certain Mn(II) concentration range (0–800 mg /L) were larger than without
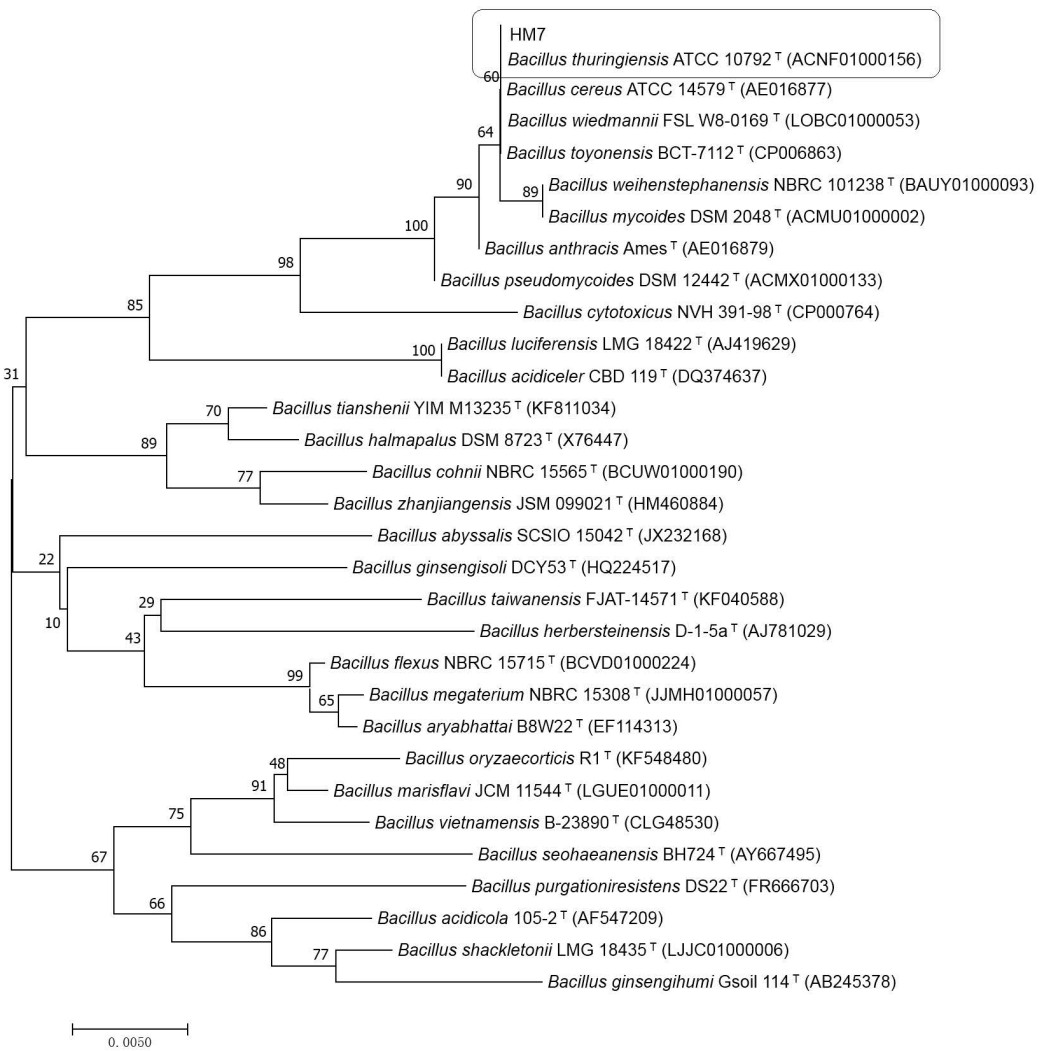

**Figure 1** **Phylogenetic relationships by a neighboring analysis of the 16S rRNA gene sequences showing the position of the strain HM7.**

Mn(II). The strain growth reached equilibrium, and the maximum $OD_{600}$ was 1.90 at 600 mg/L of Mn(II). As an essential element of biological growth, Mn(II) could promote the growth of bacteria at low Mn(II) concentration (0–800 mg/L). However, when the concentration was more than 2,000 mg/L, $OD_{600}$ had a significant downward trend and growth inhibition appeared, HM7 could hardly grow at 4,000 mg /L of Mn(II). This may be due to with increased initial Mn(II) concentration decreased the microbial activities, and finally heavy metal toxicity inhibited the growth of the bacteria (Fig. 2). Microorganisms mainly provide energy for life activities through respiration. Under the stress of high concentrations of heavy metals, oxygen consumption in respiration was significantly reduced, microbial respiration was weakened, and microbial activity was significantly inhibited (*Crane, Dighton & Barkay, 2010*). In addition, high concentrations

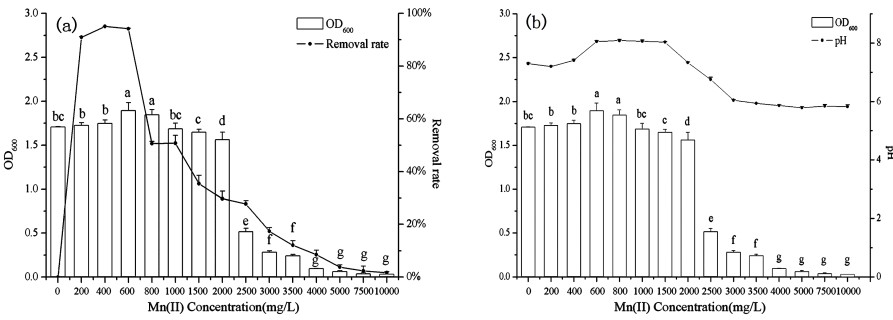

**Figure 2** The relationship between *B. thuringiensis* HM7 growth, removal rate of Mn(II) and system pH at different Mn(II) concentrations.

of Mn stimulated the production of a large number of reactive oxygen species (ROS), and ROS could oxidize cell membrane lipids, functional proteins, DNA and other biological macromolecules, resulting in cell damage (*Gong et al., 2012*).

The Mn(II) removal rate first increased and then decreased with the increased of Mn(II) concentration, the maximum removal rate was 95.04% at 400 mg/L. These results indicated that when the strain provided excessive active adsorption sites, at lower concentrations, all metal ions in the solution would interact with the binding site, so the percentage of biosorption was likely to be higher than at high ion concentrations. However, an increase in the concentration of Mn(II) might lead to competition at the adsorption sites and a decrease in microbial activity, so the biosorption rate declined (*Ling et al., 2011*; *Yu et al., 2013*).

Meanwhile, the experimental results indicated that the growth of *B. thuringiensis* HM7 ($OD_{600}$) correlated with pH ($r = 0.951$, $P < 0.05$). Before the experimental reaction, the pH in the solution was 5.6–5.8, after reacting with different concentrations of $Mn^{2+}$, the pH in the solution varied from 5.8 to 8.1. Therefore, we speculated that HM7 might release some alkaline substances during its growth or reaction with Mn(II). In a certain concentration range of Mn(II), the pH and $OD_{600}$ increased with the increase of concentration (0–600 mg/L), while, at concentrations of 600–10,000 mg/L, $OD_{600}$ and pH decreased gradually, and the solution pH from weakly alkaline to weakly acidic. This could be described that the oxidation reaction of $MnO_2$ ionic oxidation involved oxygen atoms in hydroxyl ions, and a large amount of $H^+$ in O-H was exchanged with $Mn^{2+}$ as the concentration of Mn(II) increased, resulting in low pH of the system (*Wang et al., 2015*).

### Effect of contact time on biosorption

The growth of *B. thuringiensis* HM7 and the removal rate of heavy metal increased rapidly in the beginning, but after 72 h the rate of biosorption and the growth of HM7 slowed down and reached an equilibrium with highest Mn(II) removal rate of 56.35% at 108 h (Fig. 3A). The growth of HM7 was positively correlated with the removal rate of Mn(II) ($r = 0.97$, $P < 0.05$). Due to the high affinity of the free Mn(II) binding sites on the biosorbent, the biosorption rate of Mn(II) was highest at the start time, but slowed and reached equilibrium after about 108 h. It might be described that when the increased

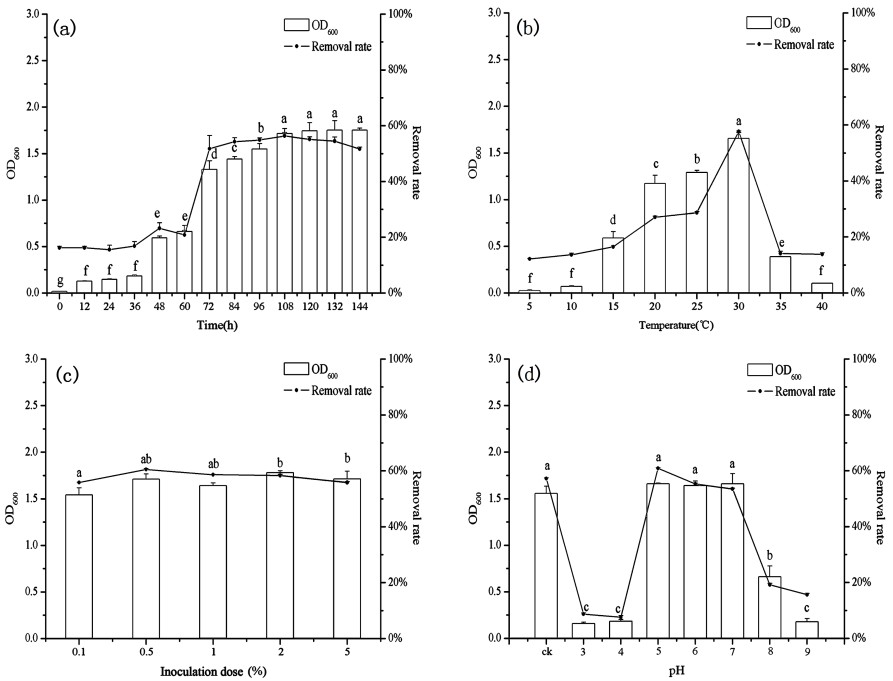

**Figure 3** The relationship between growth and removal rate of Mn(II) under different culture conditions.

biomass provided sufficient active adsorption sites on the cell surface to interact with Mn(II), and the removal rate increased. But when the biomass increased continuously, excess biomass would aggregate part of cells, and less surface area exposed, which resulted in insufficient adsorption sites for the biosorption and no excess Mn(II) adsorbed, indicating that equilibrium concentrations might be reached. In addition, the growth of HM7 was correlated with pH (S Fig. 1) ($r = 0.95$, $P < 0.05$), showing that some alkaline substances might be released when bacteria adsorbed Mn(II).

### Effect of temperature on biosorption

With increasing temperature, the removal rate and biomass were raised first and then decreased (Fig. 3B). At low temperatures (5−10 °C), *B. thuringiensis* HM7 could not grow and Mn(II) removal rates were extremely low, the removal rate and the growth of HM7 increased with increasing temperatures from 5 to 30 °C, the maximum biomass was 1.66 (OD$_{600}$) and removal rate reached 57.68% at 30 °C. When the temperature was higher than 35 °C, the adsorption capacity and growth of the strain both decreased, this may be due to the temperature was too high, and affect the activity of the extracellular polymer, thereby reducing the biosorption of metal. It was observed that extreme temperatures reduced HM7 growth and Mn(II) oxidation, this was because excessive temperatures could lead to RNA pyrolysis, protein denaturation, and microbes to stop growing or dying. On the other hand, a low temperature would reduce the metabolism of microorganisms, the fluidity of the membrane was weak and the function of the transport system was impeded, so that

nutrients could not enter the cell quickly, resulting in a low growth rate (*Hao et al., 2016*). Besides, the pH in solution correlated with the growth of HM7 and the strain might release alkaline substances (Fig. S1). With the increased of temperature, the growth of the strain increased and the pH of the corresponding solution also increased continuously ($r = 0.97$, $P < 0.05$), and this may be related to microbial adsorption of heavy metals.

### Effect of inoculation dose on biosorption

The final biomass, removal rate, and pH were stable after 72 h of culturing with different initial inoculation dose which under the same nutritional conditions (Fig. 3C and Fig. S1). As the inoculum increased, the growth of *B. thuringiensis* HM7 and the removal rate of Mn(II) did not change significantly. When the inoculation dose was increased from 0.1 mL to 0.5 mL, the maximum Mn(II) removal rate was 60.50%, which slightly higher than the others. This was because biomass provided enough active adsorption sites, Mn(II) ions can easily bind with these extra sites. It suggested that in the adsorption system depended on the metal concentration, as long as the adsorption sites on the cell surface did not reach saturation, the removal rate of heavy metals increased. When the number of cells reached a stable level, the effect on the removal rate was not obvious with continuously inoculation. In the case of certain nutrient, the effect of inoculation amount on strain growth was not significant. Therefore, in order to ensure high removal rate and cost savings, excessive inoculum should not be used (*Fan et al., 2008*). Besides, the pH of the solution with different inoculation amount did not differ significantly.

### Effect of pH on biosorption

In present tests, pH could affect the ability of *B. thuringiensis* HM7 adsorption (Fig. 3D). In an acidic condition (pH < 5), there was almost no removal rate and growth of the strain, this was due to the competitive adsorption at low pH, a large amounts of protons like $H^+$ and $H_3O^+$ increased the difficulty in binding sites of $Mn^{2+}$ to cell walls, resulting in relatively low adsorption rate. When the pH was 5.0, the highest removal rate was reached (60.88%), and the biomass achieved a maximum $OD_{600}$ of 1.66. This could be attributed to more negatively charged cells become available, promoting a greater metal uptake, and the anionic state of the functional group increased the attraction of ions to $Mn^{2+}$. However, when the pH exceeded the appropriate point (pH > 7), as it increased, biomass and removal rate would soon decrease. The reasons might be that at higher pH values, more ligands with negative charges like imidazole, carboxyl, and phosphate groups were exposed which would attract the positive charge metal ions to the cell surface. It could affect the carriers to help transport and the activity of the cell enzyme, thus the adsorption and growth of strain was restricted (*Pardo et al., 2003*; *Wang et al., 2017*). In particular, it could be seen that pH had a significant effect on the growth of the strain, the strain grew well in the weak acid environment (pH5-7). HM7 reached the maximum growth amount with the initial pH of 5.0, and the pH of the solution after the reaction raised to 7.85 from the initial state. In the case of poor growth of the HM7, the pH of the solution before and after the reaction did not change significantly (Fig. S1). This indicated that in the Mn(II) biosorption, the chemical reaction of HM7 changed the pH of the solution or secreted alkaline substances (*Liao et al., 2015*; *Wang et al., 2013*).

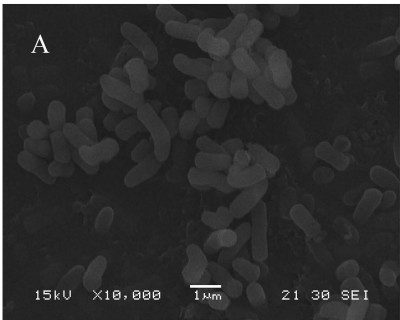
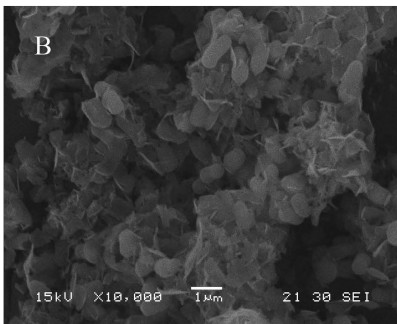

**Figure 4 SEM micrographs of HM7.** Scanning electron micrograph of HM 7 before (A) and after (B) absorbing Mn, respectively.

## SEM analysis

SEM analysis of *B. thuringiensis* HM7 to determine cell surface structure changes under 1,000 mg/L Mn(II) stress. The SEM micrographs showed that under normal growth conditions, HM7 had a morphological shape with a smooth surface and an average cell diameter of about 0.5 µm (Fig. 4A). The small size of the bacterial structure provided a large contact interface for interaction with metals during biosorption (*Zouboulis, Loukidou & Matis, 2004*). After adsorption of Mn(II), the cell surface morphology considerably changed, the length and size of the cell decreased, and the cells were irregular and cracked with the appearance of wrinkles on the surface, and there were many flocs on the surface of the strain (Fig. 4B). This might be due to the precipitation or adsorption of Mn(II) oxidation on the surface of bacterial cells and the morphological changes caused by the secretion of extracellular polymeric substances during metal biosorption (*Chen et al., 2000*).

## FTIR analysis

Biosorption depends largely on the physicochemical conditions of the solution and the functional groups of the bacterial cell active site. Therefore, to better understand the types of functional groups involved in the biosorption process, FTIR analysis was performed on HM7 (*Iqbal, Saeed & Zafar, 2009*). Intensity and peak position changed before visibly and after adsorption of Mn(II), and indicated the interaction of the relevant functional groups with Mn(II) ions in the biosorption process (Fig. 5). The Peak at 3,480 $cm^{-1}$ was shifted to 3,000 $cm^{-1}$ after adsorption corresponded to hydroxyl ($OH^{-}$) stretching vibrations (*Iqbal, Saeed & Zafar, 2009*). The significant variation of peak of 2,360 $cm^{-1}$ showed that sulfhydryl groups participated in the interaction with Mn(II) (*Dash, Mangwani & Das, 2013*). In addition, the spectrum indicated some prominent absorption peaks at 1,610, 1,500, 1,410 and 1,400 $cm^{-1}$ showed the presence of carboxyl and amide groups on the surface of bacterial cells (*Bharagava & Mishra, 2017*). Adsorption peak at 1,400 $cm^{-1}$ was protein amide I band. The peak ranges of sugars were appeared at 1,000–1,200 $cm^{-1}$ (*Deng et al., 2013*). The Mn(II) exposed strain biomass also showed two prominent peaks at 830 and 860 $cm^{-1}$, which represented CH=CH of trans-di-substituted alkenes and

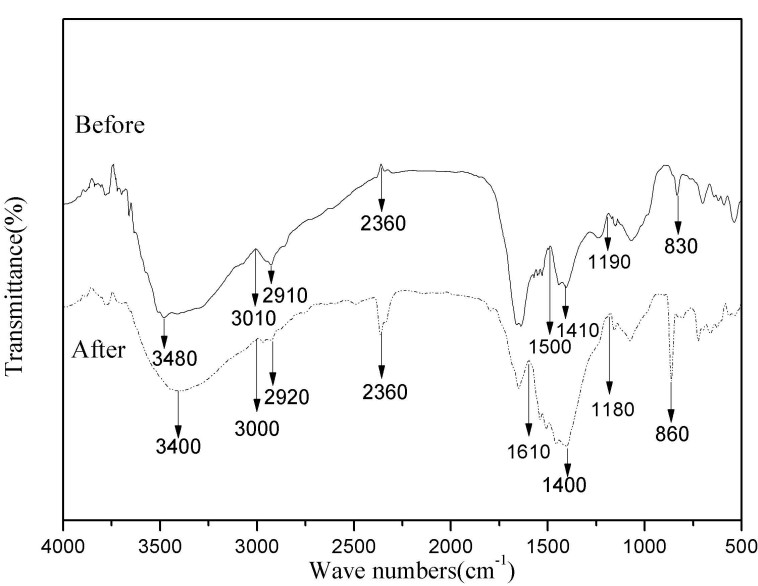

**Figure 5** FTIR of *B. thuringiensis* HM7 before and after adsorption of Mn(II).

-CH out of plane deformation interact with Mn(II) (*Bharagava & Mishra, 2017*). The results showed that the main component of HM7 surface and strain secretion was a polysaccharide compound containing a large amount of pyrrole ring and protein hydroxyl group. Comparison of FTIR spectra of HM7 before and after Mn(II) adsorption showed that the functional groups such as carboxyl, hydroxyl, sulfhydryl groups, amino, amide I and amide II bands might participate in the complexation of Mn(II).

## Characteristics of the *B. thuringiensis* HM7 in plant growth promoting and antibiotic resistance ability

It is reported that bacteria increased the biomass yield of plants and reduce the metal toxicity on metal contaminated soil by producing beneficial substances like mineral production of IAA, phosphate solubilization and siderophores (*Ma et al., 2016*; *Zhang et al., 2015*). In this study, *B. thuringiensis* HM7 might produce IAA at the level of 2.55 mg/L (Table 3), the released of IAA usually caused nutrient absorption and plant biomass to increase, thereby indirectly increased the accumulation of metal in their host (*Zhang et al., 2011*). In addition, HM7 was able to solubilize mineral phosphate and reached 8.76 mg/L, which would improve the bioavailability of rhizosphere phosphate (*Nautiyal et al., 2000*). Moreover, HM7 produced siderophores which could promote plant growth and alleviate metal toxicity of the supplement for iron nutrition in plant roots. The production of siderophores formed a strong bond with Mn(III) during Mn(II) oxygen reaction. Thus, Mn(III) was stabilized by the siderophores-Mn(III) chelate in the reaction (*Barboza, Guerra-Sá & Leão, 2016*; *Barzanti et al., 2007*).

Resistant for Amikacin, Norfloxacin, Gentamycin and Ciprofloxacin. Antibiotics and heavy metals in the environment are potential health hazards due to these properties are usually associated with transmissible plasmids, thus leading to a simultaneous selection

**Table 3** Characteristics of the *B. thuringiensis* HM7 in plant growth promoting and antibiotic resistance ability.

| Tests employed | Characteristics |
|---|---|
| **Plant growth promoting ability** | |
| IAA (mg/L) | $2.55 \pm 0.043$ |
| Phosphate Solubilization (mg/L) | $8.76 \pm 0.617$ |
| Siderophores production | + |
| **Antibiotic resistance** | |
| Cefazolin (30 mcg) | S |
| Amikacin (5 mcg) | R |
| Chloramphenicol (30 mcg) | S |
| Erythromycin (15 mcg) | S |
| Trimethoprim (5 mcg) | S |
| Norfloxacin (10 mcg) | R |
| Gentamycin (10 mcg) | R |
| Ciprofloxacin (5 mcg) | R |
| Penicillin G(10 units) | S |
| Ampicillin (10 mcg) | S |

**Notes.**
$\pm$, Standard deviation; +, Reaction; R, Resistant; S, Sensitive.

of resistance factors for metal ions and antibiotics to suit a variety of harsh environments (*Bharagava & Mishra, 2017*). Therefore, we believe that the object strain could be used as the inoculant for Mn(II) contaminated water, which may have survival advantages in the rhizosphere, as co-resistance to Mn(II) and antibiotics, and have potential in the ability to promote plant growth and alleviate metal toxicity.

# DISCUSSION

As an alternative to basic chemical methods, bioremediation has proved to be a viable strategy for Mn (II) removal. Researchers have found that the widely occurring Mn oxides in the natural environment were mainly formed through the action of microorganisms. Microorganisms could catalyze and oxidize Mn(II) to form oxides, which greatly increased the oxidation rate of Mn(II) (*Popp, Kalyanaraman & Kirk, 1990*; *Stephens, 2005*). The process of microbial accumulation was related to the metabolism of cells, because the activities of the microorganisms required the participation of metal ions, and when the cells transported these metal ions, some metal ions competed for the adsorption sites, finally formed metal compounds such as oxalic acid salts, carbonates, sulphides, hydroxides (*Kuffner et al., 2010*). Further studies have found that the adsorption capacity of Mn oxides produced by microbial transformation was generally higher than that of chemically synthesized Mn oxides (*Nelson et al., 2002*; *Tani et al., 2004*), and the primary product of bio-manganese oxidation had the activity of catalytic oxidation of Mn(II) (*Learman et al., 2011*). Therefore, the characteristics and applications of microbial removal of Mn have gradually become the current research hotspot.

Bioremediation is a significant way to solve the heavy metal pollution, because it is a cheap and environment-friendly natural processes with a high degree of public acceptance (*Hynes et al., 2008*). In recent researches, Hasan showed that *B. cereus* could effectively adsorbed Mn and Pb by interacting with environmental factors (*Hasan et al., 2016*). The research showed that *B. cereus* was a Mn redox microorganism which isolated from the biofilm of chlorinated drinking water systems (*Cerrato et al., 2010*). Therefore, *Bacillus* sp. was considered to be one of the best bacteria for adsorbing heavy metals based on former researches (*Barboza, Guerra-Sá & Leão, 2016*). *B. thuringiensis* is a Gram positive soil bacterium which plays an extremely important role in agricultural production and is often used as a biological pesticide. However, the most study of *B. thuringiensis* focused on the insecticidal specificity of toxic proteins, the structure and mechanism of toxic proteins, environmental safety and resistance, and the application in agroforestry, the researches on the tolerant and biosorption of heavy metals have not been well researched (*De Maagd, Bravo & Crickmore, 2001*; *Helgason et al., 2000*; *Nap et al., 2003*; *Sanahuja et al., 2015*; *Vachon, Laprade & Schwartz, 2012*; *Walker et al., 2003*).

In recent years, a large number of studies have shown that microorganisms have a good removal effect on heavy metals and the ability to promote plant organisms (Table 3), and HM7 exhibited high Mn(II) tolerance, good Mn(II) removal efficiency and the potential to promote plant growth. *Yan et al. (2014)* claimed that at a certain Mn(II) concentration, the removal rate of Mn(II) by *Aminobacter sp*. H1 can reach more than 90%. However, the Mn(II) tolerance of *Aminobacter sp.* H1(2650 mg/L) was much lower than that of HM7(4000 mg/L). In our previous project, three Mn(II) tolerant strains *(B. cereus* HM5, *B. thuringiensis* HM7, *R. pickettii* HM8) was isolated, the Mn adsorption capacity and characteristics of these strains were studied (*Huang et al., 2018*; *Xu et al., 2019*). The maximum survival concentration and adsorption characteristics of the above strains were different (Table S1). The results showed that the maximum adsorption capacity of *B. cereus* HM5 was 593.36 mg/L at the Mn concentration of 800 mg/L, the maximum adsorption capacity of *R. pickettii* HM8 was 1,002.83 mg/L at a Mn concentration level of 10,000 mg/L, and the maximum adsorption capacity of *B. thuringiensis* HM7 to Mn was 693.0417 mg/L at the Mn concentration of 2,500 mg/L. In addition, we figured out the ability of indoleacetic acid production, siderophore production, and solubilizing P potential. Compared with the other two strains of bacteria, the results indicated that HM7 had the best ability to produce IAA, and also had a good ability to produce iron and dissolve phosphorus. Therefore, we believe that *B. thuringiensis* HM7 has a better prospect in dealing with heavy metal pollution in the soil, especially in combination with phytoremediation.

In this study, we have demonstrated that HM7 promoted Mn(II) oxidation, which is related to the interaction of Mn(II) with bacterial components or bacterial metabolites. In the pH test, HM7 reached the maximum growth amount and removal rate when the initial pH was 5.0, and the solution pH raised from the initial state to 7.85 after the reaction. We believed that the growth of bacteria would cause changes in pH, which could affect the removal rate. Some published articles also emphasized that pH was one of the main factors involving Mn(II) oxidation (*Burger et al., 2008*; *Silva et al., 2012*). Biofilm has a significant effect on the adsorption of heavy metals, especially the oxides on the membrane, which

plays a leading role in the heavy metals biosorption. There are a large number of functional groups on the bacterial cell wall, which directly affect the biosorption of bacteria (*Wang et al., 2013*). FTIR indicated that functional groups such as carboxyl, hydroxyl, sulfhydryl groups, amino, amide I and amide II bands could participate in the complexation of Mn(II), these functional groups could be deprotonated to increase the negative charge on the cell surface and promote electrostatic interaction with the cation. Through SEM analysis, it was found that the surface of the cell changed significantly after the adsorption of Mn(II), indicating that the functional group may interact with the Mn(II) on the cell surface, and other relevant researches have also been reported (*Bharagava & Mishra, 2017*; *Oves, Khan & Zaidi, 2012*).

In recent years, many studies have demonstrated that microorganisms can secrete metal-chelators (e.g., organic acid, siderophores, and biosurfactants) phosphate solubilization and redox activity to increase bioavailability of metal by plant (*Liu et al., 2017*; *Ma et al., 2011*). Azcon et al. assayed that *B. cereus*, *Candida parapsilosis* and *Arbuscular mycorrhiza* l fungi could benefit plant growth and nutrient uptake in the heavy metal contaminated soil (*Azcón & Barea, 2010*). Sheng isolated Pb-resistant strains (*Microbacterium sp.* and *Pseudomonas fluorescens G10*) produced siderophores, indole acetic acid and carboxylate deaminase, and increased in total Pb uptake and biomass production in the bacteria-inoculated plants were obtained (*Sheng et al., 2008b*). In this study, a high Mn(II)-tolerance strain could promote metal bioavailability through metal chelating agent secretion (siderophores), phosphate solubilization and redox activity. Thus, *B. thuringiensis* HM7 could be used as a biotechnological tool to promote plant development in heavy metal contaminated environments.

## CONCLUSIONS

Mn is a significant metal that maintains a variety of biological functions, but it can be toxic in high concentrations. Therefore, Mn(II)-tolerant bacteria can be used as a safe and environmentally friendly alternative to reduce the concentration of metals at the contaminated site. In this study, a Mn(II)-tolerant strain was isolated from Mn ore soil, the strain had high tolerance to Mn(II) concentration, good Mn(II) oxidation ability, and the potential to promote plant growth. Compared the relevant researches, the results of this study are meaningful, HM7 could be used as an economical, effective and green adsorbent for the removal and recovery of heavy metals from polluted environment, but the complete mechanism underlying Mn(II) oxidation needs further investigation.

### Funding

The study was supported by the Major Science and Technology Program of Hunan Province (2017NK1014); the Forestry Science and Technology Project of Hunan Province (XLK201825, XKL201731); the Key Technology R&D Program of Hunan Province (2016TP2007, 2017TP2006); and the Key Technology R&D Program of Changsha

(kq1901145). The funders had no role in study design, data collection and analysis, decision to publish, or preparation of the manuscript.

## Grant Disclosures

The following grant information was disclosed by the authors:
Major Science and Technology Program of Hunan Province: 2017NK1014.
Forestry Science and Technology Project of Hunan Province: XLK201825, XKL201731.
Key Technology R&D Program of Hunan Province: 2016TP2007, 2017TP2006.
Key Technology R&D Program of Changsha: kq1901145.

## Competing Interests

The authors declare there are no competing interests.

## Author Contributions

- Huimin Huang conceived and designed the experiments, performed the experiments, analyzed the data, prepared figures and/or tables, and approved the final draft.
- Yunlin Zhao conceived and designed the experiments, authored or reviewed drafts of the paper, and approved the final draft.
- Zhenggang Xu conceived and designed the experiments, prepared figures and/or tables, authored or reviewed drafts of the paper, and approved the final draft.
- Yi Ding performed the experiments, analyzed the data, prepared figures and/or tables, and approved the final draft.
- Xiaomei Zhou and Meng Dong conceived and designed the experiments, prepared figures and/or tables, and approved the final draft.

## Data Availability

The sequence of 16S rDNA of HM7 is available at GenBank: MG787231.

## Supplemental Information

Supplemental information for this article can be found online at http://dx.doi.org/10.7717/peerj.8589#supplemental-information.

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
