# Peer review of "A high Mn(II)-tolerance strain, Bacillus thuringiensis HM7, isolated from manganese ore and its biosorption characteristics"

_PeerJ, doi:10.7717/peerj.8589_

## Round 0.1 · original submission · Major Revisions

Dr. Huang

Following this message are the reviews of the above-referenced manuscript. We'll be glad to consider this paper for publication after it has been revised by the reviewers' comments.

Reviewer 1 ·

Basic reporting

The study conducted for the removal of Mn(II) by microorganisms isolated from Mn(II) contaminated ore soil. The background of the study explained well in introduction part however, required the addition of some point as mentioned in general comments. Figures are relevant but the title should need to detailed information as mentioned in the general comments

Experimental design

Experimental design is explained well which come under the scope of the journal. The experimental results are reported with the instrumental supporting evidences like SEM and FTIR analysis.

Validity of the findings

The results finding is applicable for the removal of Mn(II). Their finding is elaborately discussed with comparison of other study results. Conclusion is well framed and short out the study results.

Additional comments

Introduction: Line 47-49 – Hullo et al. demonstrated……. (Tang et al. 2016). From these references, in what mechanisms used by microorganism for the removal of Mn(II). Explained it.
Line number 63-65 – Reference not given. If the data not published, please provide the methodology of tolerance test.
There is a high fluctuation of environmental condition like pH, temperature (day and night)..etc. in real field condition. The results finding from this study, is that applicable to real field especially Mn(II) contaminated soils?
Elaborate the figure 4 title and What is A and B?.

Reviewer 2 ·

Basic reporting

1. The authors should think more about the objective of this study and only elucidate relevant information that truly benefits the interested reader. I also noted that relevant information is missing in the introduction.
2. The results and discussions confirm previous observations and do not extend the available knowledge base of major output from this study. Especially, the section of discussion mostly includes general information and perspectives. Current discussion section is not worthy to interpret major output from this study. I think authors should be improved in discussion section.
3. Generally, this manuscript was properly written, but there are several spelling errors (first character upper case; singular and plural; tense) in the manuscript. It would be better to use the right spelling throughout the manuscript, and the manuscript needs to be revised by a nativate English speaker.

Experimental design

no comment

Validity of the findings

L29-38: Authors need to focus on the soil environment and bioremediation.
L 76: Can you give us some examples for the sample characteristics?
L 78-79: Before the samples were stored at −80°C, was there any pretreatment?

L84-90: Provide more detail culturing conditions for your isolation (e.g. which culture media is used in this study?).
L 97: How do you do sequencing of full length of 16S rRNA region using Illumina Hiseq? As far as I known, full length of 16S rRNA regions is impossible using current NGS technology. Please provide your strategy and methodology.
L 103-104: Provide Blast criteria
L288-293: I could not agree with this sentence. It seems to be speculated rather than interpretation from the results.
L 319-320: I could not find any data supporting this sentence. Please provide appropriate results of rephrase this sentence.
L 321: Check English.
L 318-339: I think this paragraph is no meaningful. Careful text editing will improve clarity.

·

Basic reporting

no comment

Experimental design

no comment

Validity of the findings

no comment

Additional comments

The manuscript entitled “Bacillus thuringiensis HM7 isolated from manganese ore revealed high Mn(II)-tolerance and potential for plant growth promotion” presented the effects of Bcillus thuringiensis for removing manganese at different parameters. However, lots of papers have already announced that Bacillus thuringiensis has the ability to remove heavy metals, as well as the manganese. Thus, I recommend authors should figure out the novelty of this experiment. Please provide some reasonable explanations for the manganese remove mechanisms. Moreover, there are many grammatical errors in your language expression. Hence, this manuscript should be revised very thoroughly. Besides these overall comments, I have some general comments for further improvement in the manuscript.

1. Was the manganese changed to other compounds or totally absorbed by HM7? Fig. 5b shows that the manganese was fully covered on the surface of microbes, does it mean that the activities of HM7 could be reduce by the accumulation of manganese. On the other words, the manganese was not removed, but absorbed by microbes, right?

2. The title of manuscript should be changed, because of the plant growth promotion is not the purpose of this experiment.

3. There are problems with the experimental setup. What is the other parameters when you fixed one parameter (pH, Temp, Conc….)

4. Analytic procedures was not clearly described. The OD600 is the main analysis method in this experiment, but which is not mentioned in “Materials and Methods” section.

5. Lots of conjectures were presented in the manuscript without data support. Ex: “Line 186, HM7 might release alkaline substances during growth”. Alkaline is a material which is easily to measure, authors should be better to give practical data when you make an important conclusion.

6. There is no explanation of Table 3 in the manuscript.

---

## Round 0.2 · accepted · Accept

I appreciate that authors addressed the raised comments carefully and reflected those to the revised manuscript.

I believe the manuscript is ready to be published in the journal.

Reviewer 1 ·

Basic reporting

I am satisfied the corrections made in the revised manuscript. Therefore, I can accept the manuscript without needing further review.

Experimental design

I am satisfied the corrections made in the revised manuscript. Therefore, I can accept the manuscript without needing further review.

Validity of the findings

I am satisfied the corrections made in the revised manuscript. Therefore, I can accept the manuscript without needing further review.

Additional comments

I am satisfied the corrections made in the revised manuscript. Therefore, I can accept the manuscript without needing further review.

Reviewer 2 ·

Basic reporting

There are still lack of data interpretation and analysis, while I believe that current revised manuscript could provide more insights for bioremediation research area.
So, I believe now this is an acceptable version.

Experimental design

no comments

Validity of the findings

no comments

Additional comments

I appreciate that authors addressed the raised comments carefully and reflected those to the revised manuscript.
I believe the manuscript is ready to be published in the journal.

·

Basic reporting

no comment

Experimental design

no comment

Validity of the findings

no comment

Additional comments

no comment